# Will We Unlock the Benefit of Metformin for Patients with Lung Cancer? Lessons from Current Evidence and New Hypotheses

**DOI:** 10.3390/ph15070786

**Published:** 2022-06-24

**Authors:** Pedro Barrios-Bernal, Zyanya Lucia Zatarain-Barrón, Norma Hernández-Pedro, Mario Orozco-Morales, Alejandra Olivera-Ramírez, Federico Ávila-Moreno, Ana Laura Colín-González, Andrés F. Cardona, Rafael Rosell, Oscar Arrieta

**Affiliations:** 1Personalized Medicine Laboratory, Instituto Nacional de Cancerología (INCan), Mexico City 14080, Mexico; darkpabb@hotmail.com (P.B.-B.); noryhp@yahoo.com (N.H.-P.); orozco81@hotmail.com (M.O.-M.); aleorilam@gmail.com (A.O.-R.); coling.laura@gmail.com (A.L.C.-G.); 2Thoracic Oncology Unit, Instituto Nacional de Cancerología (INCan), Mexico City 14080, Mexico; lucia.zatarain.barron@gmail.com; 3Cancer Epigenomics and Lung Diseases Laboratory 12, Biomedicine Research Unit, Facultad de Estudios Superiores (FES) Iztacala, Universidad Nacional Autónoma de México (UNAM), Mexico City 54090, Mexico; avilamore@hotmail.com; 4Research Unit, National Institute of Respiratory Diseases (INER), Ismael Cosío Villegas, Mexico City 14080, Mexico; 5Molecular Oncology and Biology Systems Research Group (FOX-G/ONCOLGroup), Universidad El Bosque, Bogotá 110121, Colombia; acardonaz@yahoo.com; 6Foundation for Clinical and Applied Cancer Research (FICMAC), Bogotá 110111, Colombia; 7Luis Carlos Sarmiento Angulo Cancer Treatment and Research Center (CTIC), Bogotá 110131, Colombia; 8Catalan Institute of Oncology, Germans Trias I Pujol Research Institute and Hospital Campus Can Ruti, 8908 Badalona, Spain; rrosell@iconcologia.net

**Keywords:** metformin, body mass index, fatty acid oxidation, PHD3, EGFR

## Abstract

Metformin has been under basic and clinical study as an oncological repurposing pharmacological agent for several years, stemming from observational studies which consistently evidenced that subjects who were treated with metformin had a reduced risk for development of cancer throughout their lives, as well as improved survival outcomes when diagnosed with neoplastic diseases. As a result, several basic science studies have attempted to dissect the relationship between metformin’s metabolic mechanism of action and antineoplastic cellular signaling pathways. Evidence in this regard was compelling enough that a myriad of randomized clinical trials was planned and conducted in order to establish the effect of metformin treatment for patients with diverse neoplasms, including lung cancer. As with most novel antineoplastic agents, early results from these studies have been mostly discouraging, though a recent analysis that incorporated body mass index may provide significant information regarding which patient subgroups might derive the most benefit from the addition of metformin to their anticancer treatment. Much in line with the current pipeline for anticancer agents, it appears that the benefit of metformin may be circumscribed to a specific patient subgroup. If so, addition of metformin to antineoplastic agents could prove one of the most cost-effective interventions proposed in the context of precision oncology. Currently published reviews mostly rely on a widely questioned mechanism of action by metformin, which fails to consider the differential effects of the drug in lean vs. obese subjects. In this review, we analyze the pre-clinical and clinical information available to date regarding the use of metformin in various subtypes of lung cancer and, further, we present evidence as to the differential metabolic effects of metformin in lean and obese subjects where, paradoxically, the obese subjects have reported more benefit with the addition of metformin treatment. The novel mechanisms of action described for this biguanide may explain the different results observed in clinical trials published in the last decade. Lastly, we present novel hypothesis regarding potential biomarkers to identify who might reap benefit from this intervention, including the role of prolyl hydroxylase domain 3 (*PHD3)* expression to modify metabolic phenotypes in malignant diseases.

## 1. Introduction

The relationship between diabetes and cancer incidence and progression has been under study for decades. Previous studies have evaluated the risk of lung cancer among patients diagnosed with diabetes, though individual studies have had conflicting heterogeneous results [1,2]. Recently, a meta-analysis from 34 studies which adjusted for smoking status identified that presence of diabetes was significantly associated with increased risk of lung cancer compared with subjects without a history of diabetes [3]. Similarly incongruously, the attempt to establish the role of the highly prescribed antidiabetic medication metformin, in the treatment of patients with cancer and with or without diabetes, has produced highly disparate results [4], and though causality may be elusive, large meta-analyses have concluded that use of metformin is associated with a reduced cancer risk, better response rate, and an increase in overall survival (OS) in patients with cancer [4]. Metformin is an appealing candidate for drug repurposing in oncology for several reasons; as one of the World Health Organization´s essential medicines, it has been massively prescribed and has a well-established safety profile [5,6]. Further, it is one of the most available and low-cost drugs in the world today; last, several basic studies have provided information which put metformin centerfold as an important metabolic regulator, inhibiting proliferative pathways [7,8,9], making this a justifiable pharmacological intervention for a disease highly characterized by a metabolic dysregulation which supports uncontrolled proliferation [10,11,12,13]. Nonetheless, use of metformin in this indication is currently not advised outside a clinical trial setting, and though some studies have shown remarkable benefits by adding metformin to standard anticancer drugs, other trials have shown a lack of effect or even deleterious outcomes, particularly in trials among patients with lung cancer [14,15,16,17,18,19,20] (Table 1). As such, it is possible that metformin efficacy in lung cancer is dependent upon specific patient and tumor characteristics, as is the case with targeted agents and immunotherapy agents which benefit only particular subgroups who present specific biomarkers (i.e., *EGFR* or *ALK* actionable mutations, as well as PD-L1 expression in tumor samples) [21,22,23,24]. More recent evidence showed that metformin efficacy also depends on the combination context of the anticancer treatment protocols assigned.

Metformin exhibits a considerable number of antitumor effects which would potentially improve lung cancer treatment, however, the current literature fails to address the differential effects of metformin in lean vs. obese subjects, as well as its newly described mechanism of action which depends on redox status of the tumor cell. Interestingly, evidence has emerged regarding the potential higher benefit of metformin´s metabolic regulation in individuals with dysregulated lipolysis, including individuals with obesity who show impaired sensitivity of adipocytes to hormonal signaling. This could result in a paradoxical effect of metformin in lean vs. obese subjects, which has been documented in individuals without type 2 diabetes mellitus (T2D) treated with metformin, though not in patients with cancer undergoing this treatment [25,26].

In this review, we summarize the evidence at the pre-clinical and clinical level which is currently available regarding the role of metformin in the treatment of NSCLC. Furthermore, we discuss the biochemical foundation behind the recent observations regarding metformin efficacy and body mass index. Lastly, we present the current perspectives and challenges towards reaching a definitive conclusion pertaining to the utility of metformin in lung cancer treatment.

## 2. Overview of Lung Cancer: Epidemiology, Treatment, and Challenges

Currently, lung cancer represents a global public health threat. Worldwide, over 2.2 million new cases are diagnosed each year, making this the second most common neoplastic disease. Additionally, approximately 1.8 million deaths occur due to lung malignancies, thus representing the first cause of cancer-related mortality worldwide [27]. Moreover, incidence continues to rise in many regions, highlighting the variations in the tobacco epidemic among developed and developing nations. Despite the concrete efforts to engage smoking-cessation policies and the advent in the past two decades of novel therapeutic strategies, the current 5-year survival rate for patients with a diagnosis of lung cancer ranges from 10–20%, though some countries have achieved better outcomes, including Japan and Israel [27].

It is now widely recognized that lung cancer is a highly complex and heterogeneous disease, and it encompasses two main subtypes, the less frequent small cell lung cancer (SCLC) and non-small cell lung cancer (NSCLC), which represents nearly 90% of lung cancer cases and is further substratified into adenocarcinoma, squamous cell carcinoma, and large cell carcinoma [28]. In patients with NSCLC, treatment strategies will reflect several aspects, including disease stage. Patients diagnosed early (stage I–II) may be candidates for surgical resection and systemic therapy. Locally advanced disease (stages IIIA–IIIB) may be resected followed by adjuvant chemotherapy or undergo concurrent chemoradiotherapy followed by durvalumab in patients with unresectable disease. In patients with metastatic NSCLC, treatment will depend on several characteristics, including patient demographics, access to therapy, and molecular stratification of the tumor according to the various druggable targets which are available [28,29,30].

Despite the current efforts to establish screening programs for high-risk individuals in several countries [30,31,32], the majority of patients worldwide are diagnosed with advanced disease, when surgery is mostly no longer feasible [28,33,34]. In addition, advances in systemic therapy such as novel targeted agents and immunotherapy, and improved techniques in radiotherapy, are still mostly circumscribed to patients with socioeconomic privileges, and as such most patients with lung cancer worldwide lack access to these advances [33]. This underscores the urgent necessity to identify new therapeutic approaches to overcome resistance mechanisms and which are available and accessible to the more than 2 million people diagnosed with lung cancer every year [35,36].

## 3. Metformin

Metformin hydrochloride is the first-line agent for T2D, a disease which currently affects over 400 million people around the globe [37]. Though currently the efficacy of metformin for treating T2D is seldomly questioned, its debut to the medical field initially spurred a widespread skepticism, mostly from the failed attempts of phenformin, a similar compound, to achieve an efficacy and safety profile adequate for clinical use [38]. Currently, metformin is being studied for its action in T2D, neurodegenerative diseases, infectious diseases, and neoplastic diseases. The pleiotropic effects of metformin are vast, and though the current literature describes significant achievements, the full extent of metformin´s mechanisms in the context of various diseases are far from established [39,40].

Metformin is a positively charged compound, therefore it requires the expression of specific proteins for its transport across biological membranes. Organic cation transporters (OCTs) have been shown to aid in the intracellular transport of metformin from plasma, particularly OCT-1, which is highly expressed by cells in organs including the liver, kidney, and intestine, and OCT-3 and MATE1 have also been shown to play a role in metformin transport. OCT expression reduction could decrease the antitumor effect of metformin [38,41,42,43]. The distribution of metformin from pharmacokinetic studies has been consistent with the tissue-specific expression of the aforementioned transporters [38]. In terms of pharmacokinetics, metformin is currently administered for T2D at doses ranging from 1000–2000 mg daily, and both an immediate release and a prolonged release formulation are available, with a similar oral bioavailability. Such doses lead to plasmatic metformin concentrations of approximately 10–40 µM, though as mentioned earlier, specific tissues will have higher concentrations from preferred uptake, such as the liver [44,45,46]. It is important to highlight, particularly regarding the effects of this review, that in non-diabetic patients, treatment with 1000 mg of metformin will achieve a plasmatic concentration of 25 µM within 3 h of administration [38].

The glucose-lowering effects of metformin were established in several basic studies, having a direct effect on the endogenous production of glucose by the liver, and this effect was mediated through a tight regulation of the gluconeogenic pathway, while glycogenolysis remained mostly undisturbed [38,47,48,49].

Most basic studies regarding the mechanism of action of this agent have reached the conclusion that metformin inhibits the mitochondrial electron transport chain (ETC), particularly by disrupting complex I, which is the site of entry for electrons moved through reduced nicotine adenine dinucleotide (NADH), [50,51,52]. The inhibition of the ETC in turn would decrease adenosine triphosphate (ATP) production, NADH oxidation, and oxygen consumption, and this would trigger a cellular response to attempt to adapt to these new bioenergetic conditions, increasing expression of glycolytic enzymes and hampering glucagon signaling. As ATP levels decrease, AMP levels rise and activate adenylate cyclase. Further, this energy shortage would also activate 5′-AMP-activated protein kinase (AMPK), an energy sensor that stimulates catabolic pathways for energy generation (Figure 1) [51]. However, it is important to highlight that, recently, this mechanism of action has been called into question, and although all the reviews currently identified build on this mode to explain the diverse effects of metformin treatment in various illnesses, it must be stated that the observations that led to establishing complex I inhibition by metformin were achieved using metformin concentrations which surpassed the pharmacological bioavailability achieved using the currently approved posology by dozens or even hundreds of times (millimolar concentrations).

Interestingly, this premise that metformin inhibits the hepatic ETC complex I is the foremost used explanation behind the potential anticancer activity of metformin treatment [53]. AMPK is a highly relevant serine/threonine protein with kinase activity which is located in the cytoplasm; this kinase has three subunits (the catalytic α-subunit, the scaffolding β-subunit, and the regulatory γ-subunit). AMPK activation occurs mainly through an increase in AMP/ADP ratio and by small molecules which mimic AMP. The AMPK complex undergoes a conformational change that improves Thr-172 phosphorylation, which in turn leads to AMPK activation, increasing catabolic pathways, and decreasing anabolic pathways [54,55].

One of the kinases which can phosphorylate and activate AMPK is liver kinase B1 (LKB1), and a study on LKB1 knockout mice showed the animals present with hyperglycemia and inactivation of AMPK and are resistant to the therapeutic effect of metformin. Some studies have shown that AMPK activation is achieved through a signaling loop by AMP binding. AMPK activation allows interaction with AXIN–LKB1, generating a ternary complex AXIN–LKB1–AMPK, which facilitates the phosphorylation of AMPK by LKB1 [54,56].

On the other hand, AMPK activation is also dependent on intracellular calcium concentration, and high calcium concentrations have been shown to promote calcium/calmodulin-dependent protein kinase kinase-β (CaMKKβ) activation, leading to AMPK phosphorylation [57,58]. Accordingly, metformin indirectly activates AMPK through the secondary activation of LKB1. LKB1 phosphorylates the catalytic α domain of AMPK and its activity is stabilized by the formation of a heterotrimeric complex with pseudo-kinase STe20-related adaptor (STRAD) and the mouse 25 protein (MO25). Once the complex is formed, LKB1 activates cell signaling pathways of cell growth regulation and catabolism [59,60]. On the other hand, LKB1 orchestrates a cellular response to energetic status and regulates the balance between catabolic and anabolic processes [61]. In this regard, AMPK activation by metformin in these studies leads to a transcriptional downregulation of gluconeogenic genes as well as the phosphorylation of acetyl-CoA carboxylase (ACC) which reduces lipogenesis and promotes hepatic mitochondrial beta-oxidation [38]. In fact, some authors have claimed that AMPK and LKB1 activation are key components for metformin to inhibit glucose production in hepatocytes and stimulate glucose uptake in skeletal muscles [62]. Further, studies have shown that metformin at clinically relevant plasma concentrations does not appreciably affect energy charge or AMP concentrations, in fact, the therapeutic effects of metformin have been observed even in the absence of AMPK expression [63]. Therefore, it is unlikely based on current literature that AMPK activation is fundamentally required for metformin action in terms of the metabolic effects of the drug [38].

Recently, an elegant series of experiments performed by Madiraju et al. have shown that the most likely primary molecular target of metformin is glycerol 3-phosphate dehydrogenase (GPD2), an enzyme necessary for glycerol to enter the gluconeogenic pathway [63]. As a three-carbon skeleton, glycerol may enter the gluconeogenic pathway depending on the cellular redox state (NADH: NAD + ratio), and metformin inhibits GPD2 activity and alters the cytosolic redox balance, suppressing the entry of redox-dependent substrates to the gluconeogenic pathway (in addition to glycerol, lactate also depends on redox state for entry to gluconeogenesis) [63]. This substrate-specific mechanism of action explains in part the low rate of hypoglycemia observed with metformin treatment, as well as the abrogation of metformin action by infusion of methylene blue (which normalizes cytosolic redox state). Moreover, this would imply that metformin may have a disproportionate benefit for individuals with dysregulated white adipose tissue lipolysis [38], and one condition in which lipolysis is dysregulated is obesity, in which an enhanced baseline lipolysis is observed, likely due to impaired sensitivity of adipocytes to hormonal signaling [25]. This novel observation would imply that the effects of metformin are different in lean and obese subjects, an observation established several years ago in the seminal studies by DeFronzo regarding the effects of metformin for muscular insulin-stimulated glucose uptake. In this study, metformin increased whole body insulin-stimulated glucose uptake, but the effect was exclusively observed in patients with T2D and obesity, highlighting the paradoxical effects of metformin in lean vs. obese individuals [25,26].

## 4. Molecular Effects of Metformin in Lung Cancer

As previously mentioned, several studies have identified an increased risk of lung cancer for patients living with diabetes, and although results have been inconsistent, some studies have further shown that patients with lung cancer and diabetes have overall a worse prognosis [64].

It is important to highlight that studies have shown inconsistent results on LC prognosis among patients with this comorbidity [64,65,66], and as such definitive conclusions have not been reached regarding this association. These contradictory results might be related to a heterogeneous population, patients with different stages or with different glycemic control, and different antidiabetic treatments.

Interestingly, among patients with T2D and cancer, studies have more consistently shown that treatment with metformin, but not with other hypoglycemic agents, has beneficial effects on survival; results further show that metformin administration is associated with a reduction in cancer risk and cancer-associated mortality [67,68]. An observational study by our group reported that in patients with NSCLC and T2D, treatment with metformin is associated with an improvement in overall survival compared with patients not treated with metformin [69]. In another study, patients with T2D and SCLC treated with metformin also showed an improvement in both overall survival and progression-free survival (PFS) compared with patients that were not treated with metformin [70]. These results suggest that metformin could improve the prognosis of patients with T2D and lung neoplasms.

In non-transformed cells, metformin reportedly works through acceleration of the glucose assimilation–consumption processes and ATP formation (a mechanism mostly thought to be mediated through the AMPK signaling pathway). In accordance with this idea, it has been suggested that the antitumor effect of metformin could also exploit the AMPK axis (Figure 1). However, the antiproliferative effects of metformin are also observed in LKB1-null melanoma cells, and therefore it must be acknowledged that the breadth of scope of the diverse anticancer pathways that could be affected by metformin treatment is still mostly unknown [71]. Further, metformin has been shown to induce cell cycle arrest and apoptosis by inhibiting mTOR activity, a process which is independent of AMPK activation [71]. Therefore, the antineoplastic effect of metformin could be mediated by AMPK-dependent mechanisms and AMPK-independent mechanisms.

### 4.1. AMPK-Dependent Mechanisms

Tumor cells have metabolic alterations that confers upon them a decreased ability to respond to an energy-deprived status, creating a metabolic vulnerability that could be used as a therapeutic approach. Several LC models have described an increase in energy generation through glycolytic processes and a decrease in ATP generation through oxidative phosphorylation (OXPHOS). Thus, the metabolic Warburg effect (WE) phenotype characteristic of LC has been associated with deregulation of signaling pathways such as the LKB1–AMPK axis.

Initially, LKB1 deficiencies were detected in Peutz–Jeghers syndrome, a disease that increases the risk of suffering pleiotropic types of cancer [72,73]. Then, it was demonstrated that several lung tumors display deletions of chromosome 19, where the LKB1/STK11 gene is located [74]. Furthermore, thirty percent of patients with advanced somatic KRAS-mutant NSCLC show LKB1 deletions that lead to a partial or complete inactivation of protein function [75]. KRAS and LKB1 co-mutation confers worse outcomes in murine models and is associated with resistance to selumetinib and docetaxel [76].

At a molecular level, KRAS and LKB1 co-mutation increases the production of reactive oxygen species (ROS) and decreases ATP generation, NADPH/NADP + ratio, and glutathione levels. This process is strongly associated with activation of the KEAP1/NRF2 pathway which increases cell survival [77]. Moreover, cancer cells deficient in LKB1 show increased autophagy activity, which provides fuel for the formation of mitochondrial energy through autophagosome formation [78].

Towards a different molecular pathway, AMPK activation upon metformin exposure contributes to the upregulation of peroxisome proliferator-activated receptor-gamma coactivator 1-alpha (PGC-1α). PGC-1α is a transcriptional coactivator responsible for mitochondrial biogenesis; it has been reported that low levels of PGC-1α are associated with worse OS [79,80]. Metformin increases PGC-1α and prevents gluconeogenesis activation [81]; altogether, these data indicate that loss of LKB1 and high expression of AMPK modify lung cell metabolism to generate energy for uncontrolled proliferation. Table 2 summarizes how metformin treatment modifies AMPK expression and activation in several LC models.

### 4.2. AMPK-Independent Mechanisms

Antitumor effects of metformin could also be mediated by AMPK-independent mechanisms. Several reports indicate that AMPK is not required for the control of glycemia. In a previous study, treatment with metformin inhibited the proliferation of LKB1- and AMPK-null cancer cells, highlighting a potential AMPK-independent mechanism of action for antiproliferative effects. An additional study corroborated these findings, reporting that metformin inhibited proliferation of LKB1-positive H1299 cells and LKB1-null H460 cells, suggesting that LKB1 is not necessary for this activity. Three reports indicated that the effect of metformin on cancer cells does not require activation of AMPK: (1) in prostate cancer cells, downregulation of AMPK did not affect metformin action and mTOR inhibition [57]; (2) the activation of AMPK was not required for the antimelanoma action of metformin and the use of an AMPK inhibitor failed to restore viability in metformin-treated cells [88]; and (3) inhibition of glucose production following treatment with metformin occurred in both AMPK- and LKB1-deficient hepatocytes [58]. These data indicate that metformin could act by an alternative pathway in specific cells and underscore the complexity of mechanisms activated by this compound.

## 5. Modifications in Cell Signaling Promoted by Metformin

### 5.1. Metformin Regulates the EGFR and IGFR Pathways

EGFR is a membrane protein that participates in cell growth and proliferation and coordinates different metabolic pathways. EGFR alterations participate in cellular metabolism through the PI3K/Akt/mTOR axis, promoting cellular glucose uptake through increased expression and translocation of GLUT1. Upper activation of the PI3K/Akt/mTOR axis promotes downregulation of thioredoxin-interacting protein, de novo pyrimidine synthesis, and a specific amino acid profile-increasing glycolytic process [89,90]. Similarly, the IGF pathway is a fundamental axis for metabolism and cell growth, and it carries out transcendent functions in apoptosis and cell proliferation. The complex has two membrane receptors (IGF-1R and IGF-2R), two ligands (IGF-1 and IGF-2), and six specific binding proteins (IGFBP-1 to IGFBP-6) [91,92].

Alterations in the IGF axis have been associated with metabolic changes in cancer cells, stimulating disease progression, and contributing to the Warburg phenotype. IGF-1 stimulates metabolic disturbances through Akt/PKB activation. IGF-1R activates mTOR, promoting protein and lipid biosynthesis, leading to accelerated glycolysis and macromolecule construction, which are important for uncontrolled cancer cell proliferation [93]. Several aberrations in the IGF-1R pathway have been reported as tumoral promoters, and they are a factor of poor prognosis. Additional studies have shown that high expression of IGF-1 and IGF-2 as well as aberrations in IGFBP-3 are associated with poor prognosis, metastases, and disease progression [92,94,95]. Both the EGFR and IGFR pathways can stimulate metabolic cell modifications in a coordinated way, acting as neoplasm promoters forming a feedback system; evidence portrays that metformin treatment may oppose some of these changes and thus exert antitumor activity (Figure 2).

Several studies have shown the effect of treatment with metformin on the IGF axis. For example, metformin reverts crizotinib resistance through IGF-1R signaling inhibition, increasing crizotinib cytotoxicity, and diminishing colony formation. Metformin attenuates PI3K/AKT and MEK/ERK signaling pathways and downregulates IGF-1R in NSCLC cell lines. In contrast, in small cell lung cancer (SCLC), metformin diminishes PI3K/AKT, but it increases MEK/ERK [96,97,98,99]. The blocking of PI3K/AKT in NSCLC prevents mTOR activation (70). In the same way, tumors with loss of tumor suppressor phosphatase and tensin homologue (PTEN) can be reduced through secondary activation of AMPK with metformin treatment [59].

Thus, metformin may have beneficial effect for NSCLC patients through its inhibition of mutant IGF-1 and EGFR signaling pathways.

### 5.2. Metformin Interacts with the SIRT1 Pathway

Another important pathway that links metabolism with cell proliferation is the one triggered by sirtuin 1 (SIRT1), which is an NAD (+)-dependent protein which has deacetylase activity. This protein acts when cells undergo metabolic stress. The role of SIRT1 in cancer and in therapy response remains a controversial topic, while it contributes to attenuation of metabolic disorders. In NSCLC cells under hypoxic conditions, SIRT1 is downregulated. The same metabolic conditions lead to AMPK inactivation due to a decrease in SIRT1/LKB1-mediated AMPK activation, thus favoring resistance to cisplatin and doxorubicin [100] (Figure 2). Interestingly, SIRT1 overexpression has been associated with development and poor prognosis of NSCLC. One study showed synergic activity when inhibiting SIRT1 expression with tenovin-6 combined with metformin, with a significant reduction in cell proliferation for the overexpressed SIRT1 cancer cells, regardless of LKB1 function [101].

There are other biomarkers still under study for their association with the aforementioned metabolic processes. This information is summarized in Table 2.

## 6. Metformin in Lung Cancer Therapy

The effects of metformin on LC as a single agent have been studied in vitro and in animal models. Metformin treatment in LC cells inhibits proliferation, induces apoptosis, and decreases colony formation. However, most of the currently published studies used metformin at millimolar concentrations, while concentrations of metformin in blood samples of patients treated with approved schemes are in the micromolar range [102]. In vivo studies using metformin in drinking water (5–250 mg/kg) demonstrated that treatment inhibits lung tumorigeneses and metastases through inhibiting both mTOR activation and phosphorylation of multiple tyrosine kinase receptors (EGFR, IGFR, and VEGFR) by AMPK-dependent and -independent mechanisms [102,103]. These data provide insight into the potential of metformin to augment the efficacy of existing lung cancer therapeutics.

### 6.1. Metformin as an Adjuvant in Lung Cancer Therapy

A combination of metformin with other therapies not only regulates the LKB1–AMPK axis, it also stimulates and regulates several cellular signaling pathways depending on the type of combination treatment. For example, metformin can inhibit hormone pathways when it is combined with hormonal treatment, inducing cell cycle arrest and apoptosis. Studies of metformin’s combination with antimetabolic drugs such as 5-fluorouracil have shown inhibition of tumor proliferation, suppression of hypoxia-inducible factor 1-α (HIF-1α), and downregulation of multidrug resistance-associated protein 1 (MRP1). In addition, when used in combination with chemotherapeutic drugs, metformin inhibits lipogenesis and cholesterol synthesis through ERCC1 downregulation, promoting tumor cell death [79,91,104,105].

#### 6.1.1. Metformin plus Platinum-Based Chemotherapy and Radiotherapy

NSCLC patients with either absence of a targeted oncogenic driver mutation or high programmed death-ligand 1 (PDL-1) expression are frequently treated with platinum-based doublet chemotherapy and bevacizumab, particularly when access to immunotherapy is lacking [106]. Median overall survival in this patient subgroup ranges from 10 to 12 months, demonstrating the need for new therapeutic opportunities.

The potential effect of combining metformin with several chemotherapeutic agents has been tested in NSCLC cell lines. Chemoresistance to cisplatin is related to ROS production, IL-6 secretion, and signal transducer and activator of transcription (STAT)-3 phosphorylation. Metformin increases the antitumor effectiveness of platinum-based chemotherapy and inhibits cisplatin-induced ROS and IL-6 secretion through modulation of the STAT3 pathway by an LKB1–AMPK-independent mechanism [107,108,109]; these studies portray the potential of combining metformin with standard first-line platinum-based chemotherapy to improve efficacy.

Several clinical trials have examined the effect of platinum-based chemotherapy combined with metformin. Initial clinical studies showed an association between metformin and decreased recurrence or progression in patients with LC and diabetes. In a retrospective study, 143 patients with LC with a pre-existing diabetes diagnosis were assessed for outcomes; the subgroup of patients treated with metformin displayed a tendency toward a better disease control rate. In addition, patients treated with insulin had a worse tumor response [110]. In the same study, metformin plus chemotherapy significantly increased PFS and overall survival (OS) (20.0 months vs. 13.1 months vs. 13.0 months, respectively; *p* = 0.007) when compared with insulin and drugs other than metformin and insulin.

An open-label phase II study enrolled patients without T2D with chemotherapy-naïve advanced or metastatic non-squamous NSCLC to receive carboplatin, paclitaxel, and bevacizumab with or without concurrent metformin (NCT01578551). The primary analysis showed that the addition of metformin improved 15% of PFS in the first year, compared with patients treated with carboplatin, paclitaxel, and bevacizumab as monotherapy. The study was stopped early due to changes in practice patterns for non-squamous treatment. However, it reported an increase in OS in patients that received metformin (15.9 vs. 13.9). The safety profile for patients undergoing the combined therapy did not highlight adverse effects attributed to metformin. On the other hand, one retrospective study reported that the combination of metformin with platinum-based chemotherapy was not associated with any survival benefit in patients with NSCLC [111,112].

In another study, 14 patients with advanced non-squamous NSCLC without LKB1/STK11 mutation were enrolled in a single-arm phase 2 study. Results demonstrated that metformin was safe and well tolerated; however, the study did not show any improvement in clinical outcomes from the intervention. A disadvantage of this trial was that it did not include a control group without metformin and results are limited by its small size [113]. A pooled analysis of the aforementioned studies revealed an improvement in overall survival (14.8 months) in patients who received metformin compared with historical controls (7.6 months) [19]. Moreover, metformin administration prevents the induction of drug resistance by cisplatin in patients with KRAS mutations and loss of LKB1 [114,115]. These results indicated that using a combination of standard first-line chemotherapy with metformin may be a promising treatment strategy for patients with NSCLC.

Recently, a hypothesis emerged suggesting that nutrient deprivation could enhance the effects of metformin during chemotherapy. Based on this idea, the phase II FAME trial was designed. Patients enrolled in this trial will receive platinum-based chemotherapy and pemetrexed with metformin alone or metformin plus a fasting-mimicking diet. It is estimated that the two treatments shall improve median progression-free survival from 7.6 months to 12 months [61].

In addition to its role in combination with chemotherapy, pre-clinical studies have shown that metformin enhances radiosensitivity through activation of ataxia–telangiectasia mutated (ATM) gene product and AMPK [116]. ATM is an important signal transduction cascade for repairing DNA damage, and establishes the degree of radiosensitivity or radioresistance. This improvement of response to radiotherapy mediated by metformin has been suggested to occur through a downregulation of the hyperactive PI3K–AKT–mTOR pathway [42].

Some retrospective cohort studies have tested the hypothesis that metformin enhances radiosensitivity in NSCLC. A study reported that patients with advanced NSCLC that received metformin plus radiotherapy (50 Gy administered in a fraction of 1.5–2.75 Gy once or twice daily), along with cisplatin, have a median overall survival of 62 months; meanwhile patients in the control group (radiotherapy plus cisplatin) only achieved a median OS of 49 months. Furthermore, distant metastasis-free survival and progression-free survival were better in the patients who received metformin [117,118].

The randomized phase II trial ALMERA compared standard radiotherapy plus concurrent chemotherapy with or without metformin in unresectable, locally advanced NSCLC patients without diabetes (NCT02115464). Treatment consisted of scaled doses of metformin orally, in the first week receiving 500 mg twice per day, in the second week 1500 mg, and in the third week 2000 mg and platinum-based chemotherapy with or without consolidation with standard radiotherapy (60–63 Gy) for six weeks. Though the study was well designed to assess the benefit of metformin in this indication, it was stopped early due to slow accrual. Among the patients who were randomized from 2014–2019 (*n* = 54), results showed that addition of metformin to chemoradiotherapy was associated with inferior efficacy and higher toxicity [16]. Another study evaluated the use of metformin and/or insulin in NSCLC patients with diabetes mellitus undergoing radiochemotherapy; the study enrolled 70 patients, and although recruitment status is complete, results have not been published thus far (NCT02109549).

Initial reporting of NRG-LU001 (NCT02186841), a randomized phase II trial of concurrent chemoradiotherapy plus metformin in advanced NSCLC, showed no significant differences in rates or grade of toxicity between groups. Metformin did not improve PFS or OS and did not alter distant metastasis. Altogether, the data indicate that the benefit of metformin in terms of radiation therapy may be circumscribed to a specific disease stage and biomarker profile (i.e., LKB1 status), warranting further research.

#### 6.1.2. Metformin plus TKIs

Several studies have evaluated the effect of adding metformin in regard to efficacy of EGFR tyrosine-kinase inhibitors (TKIs) and therapeutic resistance in LC. It has been reported that metformin can synergize with gefitinib, inhibiting cell growth, and reducing AKT/PI3K/mTOR pathway activity in LKB1 wild-type NSCLC cell lines [119]. Metformin–TKI treatment was tested in transformed TKI-resistant LC cell lines with epithelial–mesenchymal transition (EMT) patterns, and the treatment showed sensitization of TKI-resistant cell lines promoting EMT regression, increasing adhesion cadherins and inhibiting IL-6 signaling [120]. Recently, our group reported a synergic effect between metformin and afatinib, and this effect was associated with a reduction in the EGFR pathway, glycolytic, and EMT markers [9]. In contrast, another study described an increase in EGFR protein expression with metformin monotherapy, but when combined with erlotinib, metformin showed synergism and growth inhibition of EGFR wild-type LC cells [121]. In TKI-resistant cell lines with mutant EGFR, a combination of metformin with gefitinib inhibited p-IGFR and p-AKT, and showed a synergic effect in apoptosis induction [91]. It has also been observed that EGFR inhibition reactivates OXPHOS, and reverts WE in LC cells.

In another study, metformin increased BIM and BAX protein expression, exacerbating gefitinib sensitivity in LC cells both in in vitro and in vivo assays [122]. A retrospective study showed a synergic effect of metformin plus TKI in patients with T2D and LC. Two groups were evaluated, one received TKI plus metformin, and another received TKI and a different hypoglycemic agent. Patients who received metformin had a better PFS (19.0 vs. 8.0 months *p* = 0.005) and OS (32.0 vs. 23.0 months *p* = 0.002) [123]. Additionally, patients treated with metformin presented a better objective response rate (70.5% vs. 45.7% *p* = 0.017) and disease control rate (97.7 vs. 80.4 *p* = 0.009). Another important point is that metformin increased PFS independently of the first line or second line of TKI treatment [123]. Our group published the results of a phase II randomized clinical trial to compare the addition of metformin to TKI in comparison with TKI monotherapy in patients with NSCLC and EGFR mutations without a history or recent diagnosis of T2D; 139 patients were randomized, and the PFS, OS, ORR, and adverse events were evaluated. Results showed that the addition of metformin increases PFS (13.1 vs. 9.9 months *p* = 0.028) and OS (31.7 vs. 17.5 months *p* = 0.019). Metformin with EGFR-TKIs was the only factor independently associated with a better OS, decreasing the hazard of death by 48% (HR 0.52, 95% CI: 0.30–0.90; *p* = 0.035). In this study, the frequency of adverse grades 3 and 4 was similar in both groups [15]. Another similarly designed study published in 2019 examined the effects of metformin in combination with gefitinib in NSCLC patients with EGFR mutation without diabetes. The study did not identify any significant differences in median progression-free survival (10.3 months vs. 11.4 months), while median overall survival was lower in the metformin group (22.0 months vs. 27.5 months) [14]. Differences in study design (placebo controlled vs. open label), as well as important regional differences in terms of patient characteristics may explain the discrepancies in results. Recently, a subgroup analysis was published which took into consideration BMI to ascertain the benefit obtained from addition of metformin to TKI treatment in the study by Arrieta et al. This subanalysis highlights that among patients randomized to receive metformin plus TKIs, those who had a body mass index (BMI) of 24 or higher had an improved PFS and OS from the intervention independent from other factors compared with those who received TKIs without metformin (HR: 0.47; *p* = 0.003 and HR: 0.55; *p* = 0.04, respectively). This interestingly was not the case for patients with a BMI lower than 24, who did not reap benefit either in PFS or OS in this post hoc subanalysis [124]. This influence of BMI on the benefit reaped from metformin among patients with NSCLC had been previously reported in a study of 434 subjects and, in this study, patients with a BMI > 25 had improved survival outcomes when using metformin, highlighting that BMI could sensitize patients to this salutary effect [125]. The biochemical foundation for this observation may stem from a differential use of nutrients by tumor cells among subjects with higher availability of fatty acids and an altered lipid metabolism and could portray a possible and easily assessable biomarker to select patients for the intervention.

#### 6.1.3. Metformin plus Immune Checkpoint Inhibitors

Immunotherapy has demonstrated a significant benefit in NSCLC patients [126,127]. Studies in mouse models showed that the addition of metformin to nivolumab increased CD8 + tumor-infiltrating lymphocytes and decreased the production of interleukin 2 (IL-2), tumor necrosis factor (TNF), and interferon-gamma (IFN-γ) and, altogether, this protects lymphocytes from apoptosis and exhaustion. Metformin further prevents the apoptosis of CD8 + TIL in TME regardless of PD-1 or Tim-3 expression and promotes tumor cell rejection. Additionally, metformin and phenformin inhibit myeloid-derived suppressor cells (MDSCs) and, as a result, increase the antitumor activity of PD-1 blockers [128]. It is also worth mentioning a recent study that identified that obesity could shape the metabolic profile of cells (both cancer and immune cells) in the tumor microenvironment and suppress antitumor immunity. Results from this study showed that cancer cells from diverse tumors may readily adapt in obese subjects to a lipid-based metabolic phenotype, essentially “starving” lymphocytes through a strategy to improve their uptake of lipids and depriving immune cells in this nutrient competition [129]. As such, it remains to be evaluated whether metformin could be alleviating this adaptation through a differential nutrient profile which could benefit immune cells. Future studies in this regard might consider evaluating the role of PHD3, which is downregulated in tumor cells of obese subjects and is a plausible candidate for achieving this preferential metabolic profile by the neoplasm [130].

According to the data from basic studies, a phase Ib trial was developed to evaluate safety, efficacy, and pharmacokinetics of metformin and nivolumab in NSCLC. The study was divided into two parts: (A1) dose escalation of metformin started at 750 mg/day and increased up to 2250 mg/day on days 1 to 14 in each cycle. Nivolumab was administered in a dose of 3 mg/kg per day. (A2) An optimal dose of metformin was determined using the conventional 3 + 3 cohort method, and the nivolumab doses were fixed (3 mg/kg). (B) Recommended doses of metformin were evaluated in the first part, and a fixed dose of nivolumab was administered until disease progression or high grade of toxic events; also, PDL-1 is evaluated by immunohistochemistry. Currently, results from this study are pending [131]. A combination of immunotherapy and metformin could be an excellent option to avoid exhaustion of lymphocytes and to enhance response to immunotherapy, however, this hypothesis should be confirmed by future clinical trials.

The safety of metformin addition to anticancer therapy has been tested by some clinical studies, and the results showed a well-tolerated response, although common adverse events such as vomiting and diarrhea were presented, however, these events could be reversed with symptomatic medical treatment [132,133].

## 7. Hypotheses on Metformin, Obesity, and Lung Cancer

The role of metformin in treatment of lung cancer has been difficult to establish thus far. Interestingly, metformin´s salutary effect on this particular neoplasm may play out similarly to its salutary effects in individuals with or without T2D and those with or without obesity. In this regard, early studies pertaining to the use of metformin to stimulate muscular glucose uptake showed metformin increased whole-body insulin-stimulated glucose uptake in patients diagnosed with T2D who presented with obesity [38]. Moreover, this effect could not be exclusively attributed to skeletal muscle uptake, and one possibility involves glucose uptake by adipocytes, which also have insulin-dependent glucose transporters (GLUT4). Clearly, in obese individuals, higher fat mass could in turn aid in glucose clearance. The studies overall indicate that metformin´s improvement of glucose uptake is an indirect consequence of improved glucose control from decreased hepatic glucose production, however, when metformin is used for treatment of non-diabetic individuals with lung cancer many such routes will not play out similarly, since such subjects already have an adequate glycemic control. However, it may be the case that metformin could change nutrient availability, speeding up glucose clearance by peripheral muscle and adipose tissue, thus leaving the tumor cells to achieve energy requirements from other nutrient profiles. Previous studies have shown that in non-diabetic individuals, metformin treatment using a hyperinsulinemic–euglycemic clamp produced increased glucose rate of disappearance and increased glucagon levels [26]. As a result of this profile, non-diabetic patients who are obese would initiate a signaling cascade to mobilize fatty acids stored in adipose tissue cells, through glucagon activation of hormone-sensitive lipase. Many tumors have been reported to readily adapt to lipid-based metabolism, which could predominate in individuals with high availability of such stored nutrients. In a recent study by Rimgel et al. [123], the authors showed through an elegant single-cell sequencing strategy how tumor cells in obese subjects swiftly change metabolic patterns to improve fat uptake and result in tumor growth. This appears to be at least in part mediated through a decreased expression of PHD3, a prolyl hydroxylase which is relevant in regulating fatty acid oxidation; decreasing PHD3 expression by tumor cells can relieve the suppression of fatty acid transport to the mitochondria, resulting in increased beta-oxidation and ATP production.

However, if a specific tumor lacks this adaptation mechanism the outcome may be deleterious when faced with a higher availability of lipid-based nutrients for which tumor cells compete with other proliferative populations, such as CD8 + lymphocytes. Though PHD3 has not been extensively studied in lung adenocarcinomas, one recent report shows that in samples from surgically resected non-small cell lung cancer, PHD1 and PHD2 mRNA levels are decreased compared with normal tissue, but not PHD3 levels [134]. Whether PHD3 levels in tumors from lean and obese subjects are different remains to be explored, but would be plausible given the transcriptional regulation of this gene by axes such as glucagon–cAMP–PKA signaling in hepatocytes [135]. This could potentially explain the differences seen in the outcomes achieved from metformin treatment of obese vs. lean individuals and could be further explored as a biomarker to predict who could most likely reap benefit from this intervention.

## 8. Conclusions and Future Directions

Metformin is an oral antidiabetic medication that has been reported to regulate a key sensor of the energetic status of the cells and participate in the inhibition of cellular proliferation by impeding mitosis. Additionally, metformin modulates enzymes that regulate key metabolic pathways and lipid metabolism, leading to the downregulation of cell proliferation, migration, and protein synthesis. Among solid tumors, oxidative phenotype regression through metformin treatment has been broadly associated with a decrease in tumor growth. Metformin may exhibit anticancer properties through the regulation of growth factors such as EGFR and IGF. Metformin has also been shown to suppress epithelial–mesenchymal transition (EMT), impeding the spread of tumor cells. Clinically, the combination of metformin with tyrosine TKIs has been shown to improve the therapeutic response rate as well as OS in specific subgroups of NSCLC patients, particularly those with a high BMI. In addition, treatment with metformin can overcome therapeutic chemo- and TKI-resistance. However, the addition of metformin to conventional cancer therapy is still controversial, indicating that the potential benefit of this combination should be explored according to the novel data emerging regarding biomarkers and patient subgroups.

In this sense, it is necessary to find a potential biomarker to select patients that could benefit from metformin treatment. Loss of expression of LKB1 and AMPK have emerged as a response biomarker in NSCLC patients. Patients with loss of LKB1 with KRAS or EGFR mutations denote an aggressive tumor phenotype, that might be due to the loss of LKB1 as a metabolic sensor; these patients might benefit from biguanide treatment. In addition, AMPK is a potential metformin target, this receptor is lost in NSCLC, and recent reports show that activation of AMPK by metformin decreases PDL-1 levels, which in turn increases cytotoxic T cell activity against cancer cells. Lastly, the role of PHD3 which has emerged as an important regulator of the metabolic switch to lipids by specific tumors must be explored in this clinical context.

The question of which cellular processes and therapies can be potentiated by metformin still remains; therefore, it is necessary to determine the best combination strategies including metformin, the optimal dose to avoid adverse events, and lastly the potential biomarkers to implement it in the context of precision oncology.

## Figures and Tables

**Figure 1 pharmaceuticals-15-00786-f001:**
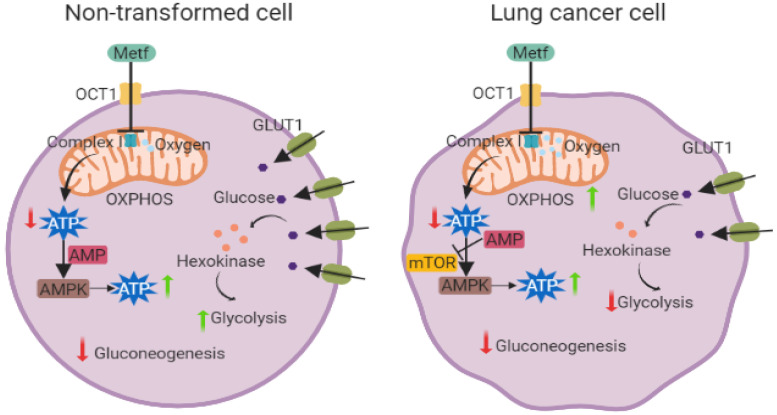
Metformin effects on non-transformed cells versus lung cancer cells. Metformin treatment has different effects on LC cells than non-transformed cells. In both cell types, metformin exerts its effect through a dysfunction of complex I of the electron transport chain, however, in non-transformed cells, metformin promotes glucose incorporation through increased expression of GLUT1 (green membrane proteins), hexokinases (orange circles), and decreasing the gluconeogenesis process. The ATP generation process is also increased through activation of AMPK and with a sustained OXPHOS. In LC cells, metformin stimulates energy generation through over-activation of OXPHOS processes associated with an increasing of AMPK pathway activity, mTOR inhibition, and a decrease in all glycolytic proteins and processes that are associated with this metabolic pathway. (Created with BioRender.com).

**Figure 2 pharmaceuticals-15-00786-f002:**
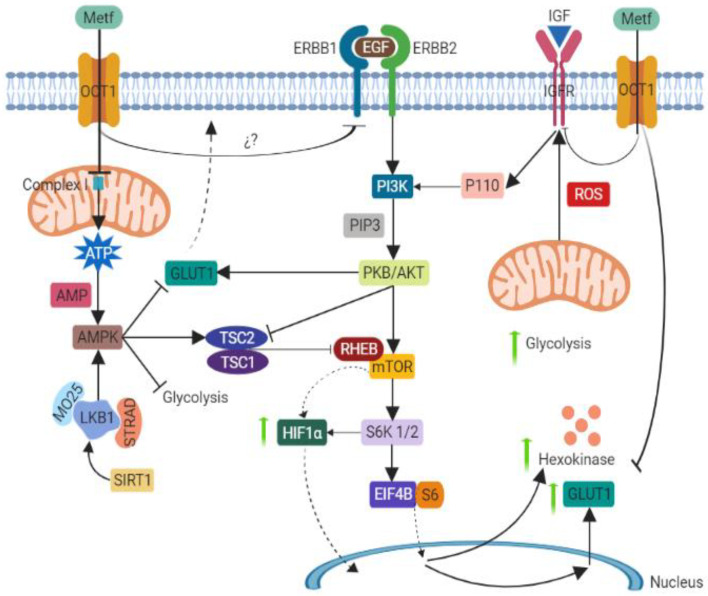
Metformin effects in the interplay between EGFR, IGFR, and AMPK pathways. Mutated EGFR and IGF pathways can increase PI3K activity, through activation of PKB/AKT, stimulating the rest of the pathway, and, also, these kinases can promote the glycolytic cell phenotype through HIF1α expression by the stimulation of mTOR and its downstream effectors. This effect promotes the translocation and increasing the expression of GLUT1 and HKs, giving cells a greater glycolytic potential. When metformin treatment is incorporated by OCTs, it can inhibit mTOR activity through AMPK activation, increasing TSC2/RHEB function, and these processes can attenuate protein synthesis that is promoted by mTOR and its effectors. LKB1 can also increase its activity through calcium-dependent processes or through the influence of SIRT1. LKB1 activation leads to increased AMPK activity associated with an inhibition of membrane EGFR and IGFR activity, and metformin has also shown direct inhibition of expression of both transmembrane receptors. (Created with BioRender.com).

**Table 1 pharmaceuticals-15-00786-t001:** Clinical trials assessing the effect of metformin for non-small cell lung cancer in patients without type 2 diabetes.

Reference	Study Design	Patients (*n*)	Primary Endpoint	Disease Stage	Treatment Arms	Metformin Dose	PFS Exp vs. Ctrl	OS (m) Exp vs. Ctrl	Considerations
[17]	Open-label, randomized phase 2 study (NRG-LU001)	*n* = 167 (*n* = 81 Ctrl; *n* = 86 Exp)	Progression-free survival at one year	III	60 Gy of radiation + concurrent weekly carboplatin and paclitaxel, followed by 2 cycles of consolidative chemotherapy every 3 weeks with or without metformin during concurrent and consolidation phases	2000 mg/day (500 mg morning; 1000 mg mid-day; 500 mg evening)	60.4% vs. 51.3%	80.2% vs. 80.8%	- Non-diabetic subjects - Patients stratified by performance status, histology, and stage - International and multi-institutional - ITT analysis - Only 39% of patients in the experimental group maintained metformin doses as indicated in the protocol - Body mass index not reported in study
[16]	Multicenter phase 2 randomized clinical trial (OCOG-ALMERA)	*n* = 54 (*n* = 28 Ctrl; *n* = 26 Exp)	Proportion of patients who experience a failure event at one year	III	Platinum-based chemotherapy, concurrent with chest radiotherapy (60–63 Gy) with or without consolidation chemotherapy with or without metformin during chemoradiotherapy and onward for 12 months	2000 mg/day	34.8% vs. 63.0%	47.4% vs. 85.2%	- Non-diabetic subjects - Trial stopped early due to slow accrual (anticipated sample = 96; actual sample = 54) - Patients were stratified for stage IIIA vs. IIIB and use of consolidation chemotherapy - G3 or higher adverse events more frequently reported in the experimental arm - Mean BMI 26.5 (Exp) 26.2 (Ctrl) - Used mostly cisplatin-based regimens - Gross tumor volume numerically larger in experimental arm - Immunotherapy (durvalumab) administered to a higher proportion of patients in control arm (15.4% vs. 25.0%)
[15]	Open-label, randomized clinical trial	*n* = 139 (*n* = 70 Ctrl; *n* = 69 Exp)	Progression-free survival	IIIB-IV	Ctrl: EGFR-TKIs (erlotinib; afatinib, gefitinib) Exp: EGFR-TKIs plus metformin	500 mg twice a day	13.1 vs. 9.9	31.7 vs. 17.5	- Non-diabetic subjects - Trial was not blinded or placebo-controlled - Further subanalysis showed that benefit from metformin was circumscribed to patients with a high body mass index
[14]	Blinded, placebo-controlled randomized clinical trial	*n*= 224 (*n* = 112 Ctrl; *n* = 112 Exp). ITT *n* = 105 Ctrl; *n* = 97 Exp	Progression-free survival at one year	IIIB-IV	Ctrl: Gefitinib 250 mg/daily + placebo Exp: Gefitinib 250 mg/daily + metformin (escalating 500 mg daily–2000 mg daily)	1000 mg BID	10.3 vs. 11.4	22.0 vs. 27.5	- Non-diabetic subjects - ITT analysis - Brain metastases were not observed in either group despite stage IIIB–IV disease in patients with *EGFR* mutations - Higher incidence of G3–4 adverse events in the metformin group (diarrhea and rash) - Body mass index not reported in study
[18]	Single-blinded phase 2 clinical trial	*n* = 15 (*n* = 1 Ctrl; *n* = 14 Exp)	Tumor metabolic response to metformin by PERCIST before definitive radiation	I-II	Stereotactic body radiotherapy to 50 Gy in 4 fractions for peripheral tumors and 70 Gy in 10 fractions for central tumors with or without 3–4 weeks of metformin	2000 mg/day (500 mg morning; 1000 mg mid-day; 500 mg evening)	Not reported	Not reported	- Subjects were randomized 6:1 to 3–4 weeks of metformin versus placebo - cT1-T2N0M0 squamous or adenocarcinoma NSCLC who were not surgical candidates - Stratification by tumor size - 57% of subjects in the experimental arm met PERCIST criteria for metabolic response. At 6 months, metformin arm had 69% metabolic response - No G3 or higher toxicities reported
[19]	Pooled analysis from two phase 2 trials		Composite progression-free survival	IV	Patients received chemotherapy (A: Carboplatin AUC 5 + pemetrexed 500 mg/m2 for 4 cycles) plus metformin 1000 mg PO BID; (B: Carboplatin AUC 6 + paclitaxel 200 mg/m2 + bevacizumab 15 mg/kd for 4-6 cycles) + metformin 1000 mg PO BID	1000 mg PO BID	6.0	14.8	- Non-diabetic subjects - Pooled data from trial A (NCT02019979, single arm) and trial B (NCT01578551, randomized 3:1) which included patients with treatment-naïve NSCLC (trial A excluded patients with *EGFR*m; trial B allowed patients with *EGFR*m who had received prior TKI therapy) - Excluded patients with brain metastases
[20]	Prospective, randomized open-label pilot study	*n* = 30 (*n* = 15 Ctrl; *n* = 15 Exp)	Objective response rate	IV	Gemcitabine/cisplatin regimen alone or with metformin	500 mg daily	5.5 vs. 5.0	12.0 vs. 6.5	- Excluded patients with diabetes and lactic acidosis - No significant increase in toxicity in experimental study arm - Non-statistically significant improvements in ORR and OS observed; no effect of metformin on PFS

**Table 2 pharmaceuticals-15-00786-t002:** Overview of studies regarding use of metformin in cancer cell lines.

Model	AMPK Modification	Treatment	Cell Metabolic Effects	Other Cell Effects	Reference
A549 and H460 cell lines	Activation	Metformin 20, 40, 80 mM	Not reported	Lung cancer cell cytotoxicity through AMPK/PKA/GSK-3β axis and mediated surviving degradation	[82]
A549 and H460 cell lines	Activation	Metformin 1mM for A549 and 2 mM for H460. Cisplatin 1 µM	Not reported	Increased apoptosis in H460 cell line in an AMPK-dependent manner	[83]
Lung cancer cells KLN205	Increased expression and activation	Metformin 5 mM in combination with 5-ALA-PDT 5 J/cm^2^	Not reported	Increased cytotoxicity, condensation of nuclear chromatin, and autophagosome formation	[84]
A549 cell line	Increased expression and activation	Metformin 0–10 mM in a combination with 2-deoxyglucose 0–2 mM	Lipid peroxidation, decreased glutathione level, super oxide dismutase and catalase activities	Enhanced cytotoxicity, DNA adduct formation, and ROS levels. Increased apoptosis and caspase-3 activity	[85]
H460 and H1299 cell lines	AMPK phosphorylation	Metformin 0–10 mM	Not reported	Cell cycle arrest, increased apoptosis, and decreased mTOR activity	[86]
A549, H460, H358 and H838 cell lines	Activation	Metformin in combination with sorafenib	Decrease in ROS production, and intracellular glutathione depletion	Antiproliferative effect associated with mTOR pathway inhibition	[87]

## Data Availability

Not applicable.

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
