# Peer review of "Will We Unlock the Benefit of Metformin for Patients with Lung Cancer? Lessons from Current Evidence and New Hypotheses"

_pharmaceuticals, 2022, doi:10.3390/ph15070786_

Round 1
Reviewer 1 Report
The subject of the manuscript is current and relevant. The authors analyze the mechanistic basis for explaining the potential advantages of using metformin as an adjuvant in the treatment of lung cancer and for using obesity as a criterion to stratify patients who may benefit most from metformin use and to explain contradictory clinical trial results. . The work is well organized, clearly written, and is a useful contribution to better understanding the mechanistic basis of metformin's effects on cancer.
Some changes are suggested that, in our opinion, may improve the manuscript.
1. Page 5, line 1: Remove "As a promising anti-cancer agent"
2. Page 5, line 5: the reference of "newly described mechanism of action ...." should be included
3. Page 5, line 9. T2D is mentioned for the first time but its definition is on the last paragraph of this page.
4. Page 6: end of second paragraph: 25 µg is not a concentration.
5. Page 8, second paragraph: It is missing the reference of MAdiraju et al.
6. Page 11, line 18: change the order of IGFR and EGFR according to the sequence used in the text (EGFR is presented in first place)
7. Page 12: On the involvement of the PI3K/AKT pathway, a putative effect of metformin on PTEN should also be included
8. Page 12, Line 60 - 67: not clear. Should be rephrased.
9. Line 13, section 6: should be reorganized. All is about the use of metformin in cancer and on the use of metformin as adjuvant. Should be " 6. Metformin in lung cancer therapy"; and 6.1.1 should be 6.1; 6.1.2 should be 6.2; 6.1.3 should be 6.3.
10. Page 13, line 121: reference 104 should be placed after ... tumor response and not only at the end of the next sentence.
12. Figures: the effects of metformin are not obvious. The arrows are too small to catch the reader's attention. Also, its meaning (color and direction) is not defined in the legends.
Author Response
Thank you for your comments. "Please see the attachment."

Reviewer 2 Report
Submitted manuscript presents reports on metformin and its therapeutic importance. The presented considerations are interesting, and allow to draw new conclusions related to the use of this drug in the treatment of cancer. At the same time, unfortunately, they do not answer the basic question about the safety of such treatment. Perhaps they only suggest a direction for further experimental research.
Other comments:
In the Tables, appropriate references to the list of References should be inserted, not the names of the authors.
Chapter 3 can be shortened.
If Figures are borrowed from another publication, in the legend of each figure should be marked from which they originate.
Author Response

(The authors gave the same response as above.)
